# Electroencephalography as a Non-Invasive Biomarker of Alzheimer’s Disease: A Forgotten Candidate to Substitute CSF Molecules?

**DOI:** 10.3390/ijms221910889

**Published:** 2021-10-08

**Authors:** Paloma Monllor, Ana Cervera-Ferri, Maria-Angeles Lloret, Daniel Esteve, Begoña Lopez, Jose-Luis Leon, Ana Lloret

**Affiliations:** 1CIBERFES, Department of Physiology, Institute INCLIVA, Faculty of Medicine, Health Research University of Valencia, Avda. Blasco Ibanez 17, 46010 Valencia, Spain; paloma.monllor@uv.es (P.M.); daniel.esteve@ext.uv.es (D.E.); 2Department of Human Anatomy and Embryology, Faculty of Medicine, University of Valencia, 46010 Valencia, Spain; ana.cervera-ferri@uv.es; 3Department of Clinical Neurophysiology, University Clinic Hospital of Valencia, Avda. Blasco Ibanez, 19, 46010 Valencia, Spain; malloretalc@gmail.com; 4Department of Neurology, University Clinic Hospital of Valencia, Avda. Blasco Ibanez, 19, 46010 Valencia, Spain; blpesquera@hotmail.com; 5Ascires Biomedical Group, Department of Neuroradiology, Hospital Clinico Universitario, 46010 Valencia, Spain; jlleong@ascires.com

**Keywords:** EEG, Alzheimer’s disease diagnosis, non-invasive biomarkers, cerebral rhythms, alpha wave, synchrony, complexity

## Abstract

Biomarkers for disease diagnosis and prognosis are crucial in clinical practice. They should be objective and quantifiable and respond to specific therapeutic interventions. Optimal biomarkers should reflect the underlying process (pathological or not), be reproducible, widely available, and allow measurements repeatedly over time. Ideally, biomarkers should also be non-invasive and cost-effective. This review aims to focus on the usefulness and limitations of electroencephalography (EEG) in the search for Alzheimer’s disease (AD) biomarkers. The main aim of this article is to review the evolution of the most used biomarkers in AD and the need for new peripheral and, ideally, non-invasive biomarkers. The characteristics of the EEG as a possible source for biomarkers will be revised, highlighting its advantages compared to the molecular markers available so far.

## 1. The Search for Biomarkers in Alzheimer’s Disease

In the field of dementia, biomarkers are used to detect the pre-symptomatic pathological changes either to evidence a pathological substrate that is related to a specific disease or to predict the progression into the spectrum of the disease. Currently, the most-used biomarkers in dementia are specific molecules that can be quantify in cerebrospinal fluid (CSF) or anomalies detected through image techniques (such as MRI or PET coupled to certain tracers) [1]. However, CSF extraction is an invasive technique, and imaging methods, although higher in spatial resolution, lack the temporal resolution needed to observe the brain in function. Some imaging techniques allow the visualization of the activation of brain areas, such as those based on the consumption of glucose, but their temporal resolution is lower than that of the EEG (seconds or minutes versus milliseconds) [2].

Since AD is the most common form of dementia, many studies have focused on specific AD biomarkers. At the end of the 20th century, the first criteria to diagnose AD were established by two independent groups (NINCDS-ADRDA and DSM-IV [3,4]). On the one hand, it was the first time that AD was recognized as a broad-spectrum disease, and the diagnosis was divided into three probability stages: probable AD, possible AD, or definitive AD. On the other hand, these criteria worked as a presumptive diagnosis, since AD definitive diagnosis still needed a post-mortem histopathological confirmation. At that time, the sensitivity and specificity were too broad, since other dementias were included as AD, blurring the final diagnosis [5,6]. 

Two pathological brain lesions characterize AD: the deposition of beta-amyloid (Aβ) plaques due to poor processing of Aβ peptide, and the formation of neurofibrillary tangles, composed mainly of abnormally phosphorylated tau (p-tau) protein [7]. With this information, new diagnosis criteria were recommended by the International Working Group for New Research Criteria for the Diagnosis of AD [8,9]. One of the latest concepts proposed by the authors was the description of AD as an evolving disease, with an earlier stage with no dementia syndrome present. This pre-dementia state could evolve to later stages when the patient is already functionally disabled, and the dementia is completely established. The aforementioned early stage is known as mild cognitive impairment (MCI), and it is characterized by memory loss symptoms that do not interfere with everyday life but are bothersome enough to seek medical advice [10,11]. 

A few years later, the National Institute on Aging and the Alzheimer’s Association (NIA-AA) [12] defined new diagnostic criteria based on biological biomarkers, as previously suggested by Dubois et al. [13]. On the whole, they proposed that AD diagnosis should be established in vivo relying on both clinical and biological parameters, requiring evidence of episodic memory impairment in addition to at least one supportive biomarker. These AD biomarkers include those that indicate Aβ deposition, such as decreased CSF Aβ or increased tracer retention on specific amyloid PET, and valid indicators of neuronal injury, such as increased CSF total tau/p-tau, hippocampal and/or medial temporal lobe atrophy on MRI, and temporal/parietal hypometabolism detected by fluorodeoxyglucose (FDG) PET. AD autosomal dominant mutation present in presenilin genes PS1, PS2, or in the amyloid precursor protein (APP) are also valid biomarkers that detect the genetic-AD form [9,13].

Additionally, another potential use of the established biomarkers is to fully identify MCI subjects that could progress to AD (MCI prodromal-to-AD), given that MCI could either remain stable or evolve to AD. Nevertheless, the MCI concept has been widely used worldwide in recent years in both clinical and research settings, and sometimes, this entity has only been referred to as a pre-AD dementia state [14]. In this line, the NIA-AA workgroup defined research criteria for MCI due to AD [10], and they proposed that the use of biomarkers could aid in identifying etiological MCI subtypes by differentiating between MCI due to AD and MCI that is unlikely to be due to AD [15,16]. 

## 2. The Need of New Peripheral Biomarkers

Nowadays, the use of biomarkers is broadly implemented in clinical practice [13,17]. The U.S. Food and Drug Administration (FDA) identified and validated them for AD [18], and today, they are routinely used by clinicians as supportive criteria for differential diagnosis. Despite their enormous impact on clinical practice, these biomarkers have important limitations: some of them (PET or MRI) are expensive to be used in routine clinical care, and their availability could be limited in some hospitals. On the other hand, CSF biomarkers are invasive and difficult for serial recordings over time [19]. Furthermore, repeated CSF extractions may cause side effects such as post-lumbar puncture headache (up to 25%), back discomfort or pain, bleeding, or even brainstem herniation [20]. 

Consequently, the scientific community has driven itself into the quest of finding new peripheral biomarkers that reflect AD’s core pathologies. Seeking a cost-effective and minimally invasive method, blood-based biomarkers associated with AD have appeared as a possible alternative. One of such proposed biomarkers is lipidic metabolite levels, which show differences between AD and age-matched controls. Additionally, a lipid combination could predict the disease progression [21]. However, it seems that lipidic metabolite concentrations are neither specific nor sensible enough to be included as an AD biomarker [22]. 

Given that a biomarker should represent the pathological process of the disease, specific AD-related proteins have also been proposed as biomarkers. Monllor et al. found a set of proteins that correlates with amyloid burden. The levels of serum clusterin, PKR, and RAGE were statistically different in the AD group compared to in controls and correlated with those of CSF Aβ42. However, the number of patients recruited was low, and confirmation in a bigger population must be performed [23]. Another approach to protein-based biomarkers was made by Nakamura and co-workers, who detected Aβ42/40 levels in plasma using immunoprecipitation coupled with mass spectrometry with very promising results [24]. Unfortunately, the method is not accessible nor easy to perform in routine clinical practice. Fully automated plasma assays have also been developed with a high value as screening test but not as a diagnostic biomarker itself [25]. Highly sensitive and specific immuno- and mass spectrometry-based assays have been recently employed to detect p-tau plasma concentrations with promising results, but more studies are necessary to endorse it as AD biomarker [26]. 

Other blood-based proposed biomarkers include plasma neurofilament light [27] and serum Glial fibrillary acidic protein [28], which reflect axonal damage and astrocytosis, respectively. Nevertheless, these proteins are not specific to AD, since they can be found in other neurodegenerative diseases, such as Huntington’s disease or frontotemporal dementia [29], or even in cognitively normal people at risk of dementia [30].

Therefore, considering the aforementioned limitations regarding current blood biomarkers, looking for alternatives is a priority in the field. From this point of view, EEG recordings could be a strong candidate as a peripheral biomarker. EEG is a direct measure of brain function with high temporal resolution and could be specific and sensible enough to detect AD-related brain changes. EEG is a non-invasive and cost-effective technique of regular use in clinical practice [19]. In the next sections, the relevance of EEG as a possible AD biomarker will be discussed.

## 3. The EEG as a Result of Brain Activity

Cognitive processing requires the temporal coordination of widely distributed neuronal populations [31]. Such interactions involve transmembrane currents, which are reflected in voltage fluctuations that can be measured invasively at the cellular level (intra- or extracellular neuronal spike recordings), at a local network level (local field potentials), or even at longer distances, at cortical levels (electrocorticograms). Electrical activity can also be recorded non-invasively by means of electrodes placed on the scalp (EEG). All these techniques allow monitoring the brain function with a high temporal resolution. Except for unit recordings, which mainly, if not only, reflect neuronal discharges, all other approaches record both neuronal discharges and other electrical sources and are represented as neural oscillations or brain rhythms [32]. However, when electrodes of bigger diameters and lower impedance are used, larger neuronal populations are recorded, and thus the spatial resolution diminishes. Consequently, the scalp EEG reflects not only the electrical activity generated by the neuronal discharges of the recorded region but also the inputs of connected regions and even distant sources whose activity is capable of travelling to the recorded sites (volume conduction). In addition, electrical noises and artifacts from many sources often interfere with the proper signals. Thus, the first relevant task when recording EEG should be the isolation and depuration of the existing signals. A limitation in dementia research is the need to reduce unnecessary movements, length, and difficulty of the experiments performed, so resting EEGs are most often chosen instead of experiments involving tasks. 

Neural oscillations have been proposed to be crucial for information processing [33]. Each cycle of the oscillation can provide a temporal window for processing the neuronal discharges occurred in this electrical context. In humans, as in other vertebrates, brain rhythms present a wide range of frequencies, from very slow oscillations occurring up to 1 in 40 s to ultrafast oscillations (up to 600 Hz), thus offering an equally wide range of temporal windows for information processing. The frequency of the oscillations relates to the extent of the network involved and is impacted by the axonal conduction velocity [34,35]: while neighboring neurons can communicate very fast through the use of low-amplitude fast oscillations, distant regions need slower oscillations, which recruit bigger neuronal populations and therefore exhibit higher amplitudes. At cortical levels, interneurons usually resonate at gamma frequencies [36,37,38,39,40]; therefore, their interactions with pyramidal cells can induce local cycles of excitation and inhibition at these frequencies [41]. Therefore, fast oscillations emerge from these local interactions. In contrast, low-frequency oscillations can be recorded at longer distances. These slow waves are travelling waves [42,43,44] and thus can be recorded far from their oscillatory generators and reflect wider network interactions at an interregional level [45]. 

Usually, neural oscillations are subdivided in frequency bands for its study. There are five traditionally defined specific frequency bands in the EEG that reflect different physiological and pathological processes: alpha, beta, delta, theta, and gamma rhythms [46]. The alpha rhythm is considered a “fast” rhythm and is the prominent EEG wave pattern of an awaken and relaxed adult with their eyes closed [47]. In general, amplitudes of alpha waves weaken when subjects open their eyes and focus on external stimuli while exercising mental effort [48]. The other “fast” rhythm of the EEG is the beta rhythm [49], which is generally of low amplitude, often enhanced during drowsiness [50], and replaces alpha activity when people become concentrated or anxious or take medications such as benzodiazepines [51]. Delta and theta rhythms are the two usually considered as “low-frequency” EEG patterns and become prominent during drowsiness and sleep in the healthy adult [50]. As people move from lighter to deeper stages of NREM sleep (prior to REM sleep), the occurrence of alpha waves diminishes and is gradually replaced by higher amplitude waves of the lower theta and delta frequencies. Additionally, neocortical slow oscillations (<1.5 Hz) can be recorded, although they usually are discarded from the analyses due to the influence of artifacts [52]. Finally, gamma oscillations are involved in attention, perception, and memory and have been proposed as biomarkers of cognitive decline in the elderly [53]. Due to computational limitations, older studies restricted the EEG analysis to these, or even narrower, frequency ranges (see Table 1). More recently, other oscillations, such as spindles and ripples, are also included in the analyses. 

When reviewing EEG studies in the literature, the first thing to note is the lack of standardized classifications of the different rhythms and their further subdivisions (i.e., low and high theta; alpha 1, alpha 2, beta 1, beta 2; and low, mid, and high gamma) by frequency range. Thus, it is highly recommended to consider the exact frequencies analyzed instead of the band names used in each study (Table 1). Usually, but not always, frequency ranges are classified as follows: delta (from 1 to 3 or 4 Hz), theta (from 3 or 4 to 8 Hz), alpha (from 8 to 12 or 13 Hz), beta (from 12 to 30 Hz), and gamma (>30 Hz). Furthermore, these frequency ranges could differ between humans and animal models, where theta can include a wider range of frequencies (from 2.5 in anesthetized animals to 12 Hz) and thus overlaps with delta and alpha bands. 

Additionally, it is important to consider not only the rhythms themselves, but also the brain area where these oscillations are recorded. For example, alpha rhythms correspond to different neural correlates when occurring in the occipital or in the temporal cortex, since they reflect the activity of different neuronal populations. 

Finally, although the EEG provides immediate reports of the brain’s electrical activity, this technique is usually restricted in clinical practice to study epilepsy and sleep disorders and is sometimes considered a subjective technique [54]. This last consideration is due to excessive *de visu* interpretation of data, which could lead to a high variability in results. In order to make an objective analysis, the use of numerical analysis of the signals (quantitative EEG or qEEG) is strongly encouraged [55]. The focus of this review lies on the results of qEEG analysis conducted on AD patients. These analyses mainly involve spectral decomposition of the signals into their frequency components and the study of their synchronization and complexity (see Table 1). 

## 4. EEG Anomalies in AD

Healthy aging is the cause of changes in the activity of the brain that, accordingly, are reflected in EEG recordings. In summary, they include a reduction in the amplitude of alpha activity (8–13 Hz), the slowing of the background activity, and an increase in delta (1–4 Hz) and theta (4–8 Hz) power [56]. However, all these physiological EEG variations are pathologically exacerbated in AD patients. Next, we will analyze the main EEG anomalies found in Alzheimer’s patients. They can be grouped into three categories: change in frequency pattern, reduction in the complexity of EEG signals, and perturbation in EEG synchrony. 

### 4.1. Change of Frequency Pattern on EEG

Most EEG studies devoted to the diagnosis of AD have been based on the spectral decomposition of the scalp signals in the resting state with closed eyes. Regarding spectral decomposition, it is important to consider the type of analysis performed in different studies, since the results can vary depending on these methods. For instance, different spectral parameters such as peak frequency, mean frequency, absolute or relative power are analyzed. 

It is important to distinguish changes that appear in MCI and AD from those due to normal aging. Babiloni et al. used low-resolution brain electromagnetic tomography (LORETA) analysis in a large sample of healthy elderly and young individuals. By combining imaging and qEEG analysis, this technique estimates the location of electrical activity generators in the brain [57], which is also named as the inverse problem [58]. They confirmed that alpha rhythm in posterior cortical regions decreases in magnitude during physiological aging, and this correlates with the global cognitive level [58]. However, many studies report that changes in EEG recordings of both MCI and AD suffer a change in pattern compared to age-matched controls: alpha and beta rhythms usually decrease, while there is a general increase in delta and theta oscillations [59,60,61,62,63,64,65,66,67,68,69,70]. Due to these findings, these changes have been traditionally described as “a slowing of the EEG”, which is often correlated with a decreased state of arousal and cognitive processing (Table 1). In order to consider this change in raw EEG frequencies as an AD biomarker, it should have a clear biological basis specifically related to the development of the disease. Indeed, it seems that EEG slowing could be linked with the atrophy reported in AD patients of basal forebrain cholinergic neurons, which innervate the neocortex and hippocampus, among others [71]. EEG slowing has also been correlated with the cognitive status of AD patients [72] and with the Folstein Mini-Mental Score (MMSE) [73,74] and global deterioration score [75]. Importantly, this slowing of EEG correlates with gold standard biomarkers of AD, such as Fluorodeoxyglucose-PET images [76] and phosphorylated tau and/or Aβ42 measured in CSF [77,78,79,80].

Early studies already found that dominant occipital frequency decreases as the neurodegeneration progresses, with demented patients exhibiting a peak frequency within the theta range (5–8 Hz), together with generalized EEG slowing over most brain regions [59,60]. Additionally, occipital EEGs of mild AD patients show a higher fraction of total power in the theta band and a lower fraction of total power in the beta band than in healthy controls, making the mean frequency higher in age-matched controls than in mild AD [61]. In this line, EEG slowing has also been described in MCI patients compared to control subjects [81]. 

This increase in slow waves, specifically in theta rhythm, occurs at the initial stages of the AD. The theta power increase coincides with the earlier signs of cognitive decline [82]. Moreover, theta relative power (measured as % of the 4–13 Hz band) is higher in AD than in MCI and higher in MCI than in healthy controls and is related to decreased performance in all cognitive domains. This study also found that theta absolute power is higher in AD than in healthy controls. Longitudinal studies corroborate the relevance of the anomalies in the EEG as indicators of the disease progression [82]. In this line, a study conducted by Prichep et al. proposed that theta power and mean frequency are relevant components while considering the possible progression from MCI to AD [83]. 

Moreover, the reduction in alpha activity shows a correlation with the severity of the disease and the cognitive deficits [63,84]. Combining EEG-LORETA and MRI studies, Babiloni and colleagues found a significant linear correlation between hippocampal volume and the magnitude of alpha1 (8–10.5 Hz) sources in the parietal, occipital, and temporal areas, and between progressive atrophy of hippocampus and decreased cortical alpha power, in a continuum from MCI to AD [85]. Thus, alpha power may be used as an indicator of the cognitive impairment degree. Later, the same group, in a year-long follow-up study, suggested that cortical sources of different EEG rhythms are sensitive towards the progression of early-stage AD [62]. Mild AD was characterized by increased power of widespread delta sources as well as decreased power of widespread alpha and posterior beta 1 (13–20 Hz) sources [86].

The combination of neuroimaging and quantitative EEG proves a useful tool for differential diagnosis with other dementias. Gasser and colleagues found that this combination had a greater differential diagnostic contribution than clinical symptoms and neuropsychology [55]. They observed that delta power was higher in mixed dementia than in AD, still being higher in both cases than in healthy age-matched controls. Additionally, high frequency power was nearly normal in mixed dementia, but decreased in AD. 

The possible relation between EEG slowing and the currently used biomarkers may be an interesting topic to address, on the one hand, to improve the knowledge and feasibility of EEG as possible biomarker, and on the other hand, to assess the possibility that EEG changes may reflect the extent of pathological degeneration. In this line, a study reported an inverse correlation between EEG slowing and CSF tau levels in advanced AD patients [87]. Another study reported that tau, p-tau, and p-tau/Aβ42 ratio showed a correlation with relative theta power in cognitively normal old subjects who developed memory complaints throughout the follow-up [77]. It was also reported that decreases in cognitive speed appeared to correlate with increased theta power. Together, these results suggest that CSF biomarkers may be related to EEG theta activity, and this feature of EEG could be potentially used to assess early abnormal degenerative changes in the brain.

### 4.2. Complexity Reduction in EEG Signals

EEG signals are non-stationary, complex, and non-linear signals [88] caused by the interaction of different sources (oscillators) [46]. Therefore, the EEG exhibits complex behaviors which cannot be linearly analyzed, and consequently, non-linear analyses have been introduced to the study of EEG signals to quantify this complexity [89]. In some diseases, the combination of linear and non-linear analyses can improve the accuracy EEG-derived biomarkers [88]. Non-linear analyses provide a way to relate the complexity of brain signals to functional aspects of the neural networks, such as the integrity of the neural connectivity or the variety of generators contributing to a given oscillatory signal [90]. In consequence, they prove useful in the search for biomarkers of different neurological and psychiatric conditions [91,92,93]. Additionally, these measurements have recently been successfully combined with machine learning techniques in order to discriminate cognitive performance [94]. Theoretically, higher complexity measures in the EEG signal are thought to reflect an integration of information among segregated groups of neurons performing different processing tasks and at different spatial scales, whereas reduced complexity reflects a lower degree of information exchange. Thus, complexity analyses provide a measure of the amount of information that is integrated within a neural system [95]. 

The reduction in EEG complexity in AD patients can therefore be interpreted as an alteration in the information exchange, as reported by numerous studies [96,97]. For example, Jeong et al. [98] used cross-mutual information and auto-mutual information between EEG electrodes as a measure of complexity in EEG activity. These parameters measure transmission among different cortical areas. By studying both AD and healthy control subjects, they concluded that auto-mutual information declines in the EEGs of AD subjects, suggesting a less complex activity on their EEG compared to normal controls, as supported by other studies [99,100]. Moreover, the authors reported that AD patients show a reduction in the cross-mutual information compared to cognitively normal subjects. In particular, the reduction was higher between interhemispheric (distant), compared to local transmission of information [98], a result possibly related to the neuronal loss happening in AD. 

Other techniques have been used to estimate EEG complexity in the context of AD. For example, approximate entropy is a non-linear statistic that can be used to quantify the irregularity of a time series [101]. Several studies have shown a decreased irregularity in the EEGs of AD patients compared to those from age-matched controls [102,103].

To summarize, there is evidence that EEGs of AD patients seem to be more regular (that is, less complex) than EEGs of age-matched control subjects. This feature could be related to the increase in slow EEG rhythms explained in the previous section, since the slowing of EEG would increase the signal regularity making it less complex [104].

### 4.3. Perturbation in EEG Synchrony and Directionality

Synchronization between neuronal populations is relevant for to the interaction among neural networks. EEG synchrony refers to the adjustment of different neural oscillations. Two different signals become coupled when both begin oscillating at the same frequency, become phase-locked, experience phase-amplitude coupling, or modulate their amplitudes together (amplitude–amplitude coupling) [105]. Different measures are used to quantify the synchronicity of neural activity. One of them is the measurement of spectral coherence, which corresponds to the spectral covariance of the activity between two electrode locations. 

It has been reported that resting state EEG coherence is reduced in AD patients compared to cognitively healthy but depressed patients [106]. Furthermore, EEG coherence in AD subjects showed statistically significant differences compared to healthy controls [107,108,109,110,111,112,113]. Curiously, Sankari et al. [108] found a significant decrease in EEG coherence in the delta band measured in temporal regions compared to healthy controls. In the same study, the parietal and central regions showed, instead, a reduction in the EEG coherence in the theta and alpha bands. Thus, slow EEG frequencies in subjects with AD are increased but with an altered spatial organization. Another study [114] reported that AD subjects showed a decrease in alpha wave coherence in temporo-parieto-occipital areas but an increase in delta wave coherence between frontal and posterior regions. These two events can be related, respectively, to the alterations in cortico–cortical connections and to the degeneration of subcortical structures such as the thalamus [71,98,115].

Using a multiple logistic regression, Prichep et al. proved that theta power (3.5–7.5 Hz) mean frequency and interhemispheric coherence predicted the decline from MCI to AD at long term, with an overall predictive accuracy of about 90% [116]. 

Regarding non-resting state, a study showed that neural synchrony in alpha2 (10–12 Hz) and beta (12–30 Hz) bands in AD patients was reduced during a working memory task, compared to control subjects. Additionally, a non-linear measure, synchronization likelihood in the alpha band (8–10 Hz), was significantly higher in MCI compared to the control subjects [111]. Another study showed that, under intermittent photic stimulation, AD patients presented a reduced coherence, regardless the stimulus frequency [117].

Additionally, the directionality of the functional coupling can provide relevant information. In this sense, using the directed transfer function (DTF), Babiloni et al. showed that parietal to frontal direction of the information flux within EEG functional coupling of theta (4–8 Hz), alpha 1 (8–10 Hz), and alpha 2 (10–12 Hz) were stronger in healthy controls compared to those in both MCI or AD [118]. In their study, they did not find any differences in the directional flow within inter-hemispheric EEG functional coupling. Later, with the aim of optimizing the selection of frequency bands, Gallego-Jutglà et al. found that the frequency range of 5–6 Hz (within the standard theta band) offered the best accuracy for diagnosing AD for DTF Granger causality [119]. 

Delbeuck et al. [120] defend that the pathophysiology underlying this synchrony loss is the degenerative processes caused by the disease. Neurofibrillary tangles and amyloid plaques would physically interrupt the electricity flow between the long cortico-cortical tracts, leading to a neocortical disconnection between neurons. Other authors defend that the perturbation in EEG synchrony has a mixed origin that involve a loss of cortical neurons in combination with reduced cholinergic activity in the cortex [98]. 

Taking into account the previously discussed hypothesis of the severe impairment of basal forebrain neurons, all three EEG features in AD, namely changes in frequency patterns, complexity, and synchrony measurements, may be the result of the loss of neurons, the altered anatomical structure of the neuronal tracts, plus the altered release of neurotransmitters, all this resulting in impairments in the neural activity [121].

## 5. Discussion

Biomarkers in AD provide a useful tool for clinical practice and research. In contrast to the stablished AD biomarkers, EEG recordings are minimally invasive, cost-effective, and can be performed with portable systems, which facilitates access to patients when necessary [122]. With all these features, EEG analysis could be a source of good candidates as peripheral biomarkers. 

The pathophysiology underlying the alterations in EEG described above is not fully understood. However, plaques and tangles could clearly provoke the interruption of the information flow along corticocortical tracts, leading to a neocortical disconnection between neurons. It has been also described that Aβ plaques could have a toxic effect in their surrounded inhibitory and excitatory neurons [123,124], thus disrupting neuronal networks [125]. Moreover, prior to amyloid plaque formation, oligomeric Aβ peptide causes the hyperactivity of hippocampal neurons and network hypersynchrony [126]. Furthermore, p-tau also changes the normal network functioning by depressing synapse efficiency and quantity. Moreover, a process of demyelination has been demonstrated very early in AD, even before atrophy of grey matter [127]. Changes in myelin thickness could therefore influence network synchronization, resulting in disrupted oscillations [128]. In this regard, for all the above, the proposal that AD is a disconnection syndrome could be justified [120,129,130]. 

Moreover, AD is not a stationary disorder. Instead, it is considered as a long-evolution disease that progresses over years [131,132]. In this progression, pathophysiological processes will have direct consequences on neuronal transmission that can be detected by EEG. Thus, EEG could reveal differences throughout the AD continuum. For instance, studies comparing MCI patients and cognitively healthy subjects show a reduction in alpha interhemispheric coupling [133] and abnormalities in alpha (8–12 Hz) band power and synchronization at resting state [134]. Furthermore, EEG has ben also studied as a tool to predict MCI conversion into AD [135,136]. Vecchio et al. [135] showed that EEG connectivity analysis, combined with a neuropsychological MCI pattern and ApoE genotyping, reached high sensitivity/specificity and accurate classification on an individual basis (>0.97 of AUC), helping to determine the risk of the progression to AD in MCI patients. Jelic et al. [136] found that the most important predictors were alpha and theta relative power, as well as mean frequency from left temporo-occipital derivation (T5-O1), which classified 85% of MCI subjects correctly.

Baker et al. were able to distinguish two different profiles inside the MCI group: one with EEG beta power profiles similar to AD patients, and one similar to controls [81]. In this line, del Val et al. [137] showed that amnestic MCI individuals with lower capacity to recruit alpha oscillatory cortical networks developed dementia in a two-year follow-up study compared to healthy controls.

Furthermore, different studies have shown the ability to predict the progression of the disease combining a variety of EEG measurements. Poil et al. used an integrative approach to improve the prediction of progression from MCI to AD, extracting multiple biomarkers, including spatial, temporal, and spectral parameters from the EEG, and selecting those that better classified the groups [138]. They found that MCI patients who progress to AD showed differences in beta (13–30 Hz) peak width, beta bandwidth, beta amplitude range, beta amplitude correlation, alpha relative power, and alpha/theta power; combining these six EEG biomarkers into a diagnostic index, they are able to predict the conversion to AD with a sensitivity of 88% and specificity of 82%.

There are also studies that focus on AD diagnosis; for example, Henderson et al. used a fractal dimension-based method for analyzing the EEG from both AD subjects and cognitively healthy controls [139]. Even with the small sample size, they showed that a single fractal measure could discriminate between AD subjects and controls with a 67% sensitivity and a specificity of 99.9%. Similarly, in a recent study, Ge et al. [140] proposed a framework that systematically discriminates among AD patients and age-matched controls based on EEG signal processing. Combining several EEG features, they obtained a ROC curve with an area under the curve of 97.92 ± 1.66 (%). Many studies show that the combination of different biomarkers improves accuracy, sensitivity, and specificity for AD diagnosis [23,141,142]. This matches with the idea defended by Dauwels et al. [143] that the combination of various EEG features could be a good approach to obtain a diagnostic tool for AD. 

## 6. Conclusions

EEG is a non-invasive, low-cost technique that constitute a good alternative to the gold standard biomarkers for AD. Dellabadia et al. found that MRI and PET cost were, respectively, three and six times higher than that of EEG [144]. Furthermore, EEG does not need to be performed in hospitals, as it can be done in primary care centers or even in ambulatory environments, making it a more flexible alternative for this vulnerable population [145]. 

Although EEG recordings in the elderly are different compared to those in younger people, the process of dementia leads to the appearance of pathological changes in the EEG that are clearly distinguishable from those of aged-matched controls. Three features of the EEG have been related to AD: a slowing of EEG, namely a reduction in fast and increase in slow frequential components, a perturbation in EEG synchrony, and a reduction in its complexity measures. Particularly, the increase in frontal delta and theta rhythm and the decrease in posterior alpha rhythms seem to reflect the AD pathophysiology. 

However, to use EEG parameters as biomarkers for AD, there is still a need to overcome the current differences in the methodological approaches for data acquisition and signal analysis. Currently, the International Federation of Clinical Neurophysiology is working on establishing unanimous recommendations for the topographic and frequency analysis of resting state EEGs. These recommendations include: how to record EEGs (environment, montage, settings); digital storage of EEG and control data; extraction of synchronization, connectivity, and topographic features; and statistical analysis and neurophysiological interpretation of those EEG features [146]. Setting up an international working group specialized in dementias could be an interesting and useful idea in order to standardize the use of EEG in the diagnosis of EA.

Yet, considering the complexity of AD pathophysiology, the current knowledge suggest that a set of EEG-derived biomarkers could reach an acceptable accuracy, sensitivity, and specificity for the diagnosis of AD. Moreover, a well-defined combination of EEG features could potentially be used to estimate the severity and the progression of the disorder.

## Figures and Tables

**Table 1 ijms-22-10889-t001:** Main studies addressing EEG analysis in Alzheimer’s disease with their main findings and listed chronologically.

Reference	Subjects	Frequency bands	Analysis	Hallmarks
Prinz et al., 1982 [60]	22 HC (11male, 11 female), 18 mild AD (9 male, 9 female), 16 moderate AD (10 male, 6 female) and severe AD (10 male)		Spectral analysis; dominant occipital frequency (DOF).	DOF decreases inversely with the progression of the neurodegeneration. A discriminant analysis DOF and sleep variables correctly classified 71% of the subjects.
Coben et al., 1983 [61]	40 mild AD patients and40 HC; Age range: 64.2–82.5; 21 female/19 male	Delta: 1–3 Hz (only 3 Hz for power), theta: 4–7 Hz (5–7 Hz for power), alpha: 7.75–13.50 Hz Beta: 14–20 Hz.	Spectral analysis of spontaneous occipital EEG (Fraction of total power in the 3–20 Hz or 5–20 Hz band; average mean frequency; alpha index (percent time alpha rhythm).	Fraction of total power in theta mild AD > HC; fraction of total power in beta mild AD < HC in occipital EEGs; average mean frequency HC > mild AD.
Rae-Grant et al, 1987 [64]	139 AD patients (69 male, 70 female)and 148 HC; Age range: 50–90		Longitudinal (4 years) correlated with standardized test and autopsy; DSM.	Slowing of the background activity (delta and theta increases), superimposition of focal abnormalities, spikes, sharp waves, asymmetries and triphasic waves. Excessive delta and triphasic waves only in AD. More severe EEG abnormalities (excessive delta) correlated with hippocampal neuron density and less with granulovacuolar ratio in autopsies.
Dierks et al., 1993 [66]	35 HC and 35 probable AD patients (age range 45–85)	Delta (1.0–3.5 Hz), theta (4.0–7.5 Hz), alpha (8.0–11.5 Hz), beta1, (12.0–15.5 Hz), beta2 (16.0–19.5 Hz), and beta3, (20.0–23.5 Hz).	Spectral analysis–dipole approximation; FFT power.	AD patients showed a shift of alpha and beta activity toward frontal brain regions which correlate with the degree of dementia. AD patients had higher power delta and theta, correlating with the severity of dementia, and lower power in the alpha and beta range. Theta is the most sensitive band. FFT dipole approximation results are constant.
Besthorn et al., 1994 [109]	50 AD patients (18 possible AD and 32 probable AD) and 42 HC	Delta 1.5–3.5 Hz, theta 3.5–7.5 Hz, alpha1 7.5–9.5 Hz, alpha2 9.5–12.5 Hz, beta1 12.5–17.5 Hz, beta2 17.5–25.0 Hz.	Spatially averaged spectral coherence between individual electrodes and all neighboring electrodes, frequency bands.	AD showed decreased coherence, mostly in the frontal and central derivations of the theta, alpha and beta frequency bands. A discriminant analysis had a 76% accuracy of prediction (patient or control) using Cz-alpha1, Pz-beta2, C3-beta1, C3-alpha1, and T4-beta 2.
Locatelli et al., 1998 [114]	10 mild or moderate AD (age range 53–77) and 10 HC	Delta 0.5–4 Hz, theta 4–8 Hz, alpha 8–12 Hz, beta 12–30 Hz; frequency resolution of 0.5 Hz.	Standard tests, imaging (CT or MRI). Mean spectral coherence of 50 artifact-free 1 s duration epochs. Coherence was calculated as the average of coherence values between electrodes.	Decrease in alpha band coherence in AD, in temporo-parieto-occipital areas, more evident in severe cognitive impairment. Delta coherence increased in a few patients between frontal and posterior regions. Trend towards a reduction in coherence in the temporo-parieto-occipital regions for the theta and beta bands in the AD. Interhemispheric delta and theta coherences tended to increase in all the analyzed pairs of electrodes (exception: F7–F8 and T5–T6). In these regions, and in the beta band, a coherence decrease was present in AD.
Claus et al., 1999 [65]	86 probable AD (49 male, 37 female) and 49 HC		Visual inspection with Grand Total EEG (GTE) score. Standardized tests for cognition.	Abnormalities in the visual inspection of the EEG can increase the diagnostic of mild AD in in diagnostic doubt (with low sensitivity). Frequency of rhythmic background activity, diffuse slow activity, and reactivity of the rhythmic background activity were statistically significant related to the diagnosis.
Kowalski et al., 2001 [67]	95 probable AD (mild, marked, and severe dementia); 75 female, 20 male	Theta 6–7 Hz; 5–7 Hz; 4–7 Hz; delta 3 Hz; slow waves: delta and theta.	Standardized test. Descriptive (visual) analysis. Eight-degree scale according to the background activity, presence and amount of theta and delta waves, focal changes, lateralization of focal changes, synchronization, and presence of sharp and spike waves.	Significant correlation between the degree of EEG abnormalities and cognitive impairment. No correlation between delta waves and MMSE nor GDS. No association between duration of the disease and degree of EEG abnormalities
Stam et al., 2003 [110]	10 AD (2 male, 8 females; age range 59–86), 17 MCI (8 male, 9 females; age range 62–88) and 20 with subjective memory complaints (SC) (11 male, 9 females; age range: 51–89)	2–6 Hz, 6–10 Hz, 10–14 Hz, 14–18 Hz, 18–22 Hz, and 22–50 Hz (based upon the suggestions of Leuchter et al., 1993).	Standard tests and imaging for diagnosis. Synchronization likelihood (coherence measure), comparing each channel with all other channels.	Synchronization likelihood decreased in the 14–18 Hz and 18–22 Hz bands in AD compared with both MCI subjects and SC. Lower beta band synchronization correlated with lower MMSE scores.
Pijnenburg et al., 2004 [111]	14 AD (7 male, 7 female), 11 MCI (10 female, 1 male) and 14 (8 male, 6 female) with subjective memory complaints (SC) (SC were younger)	0.5–4 Hz, 4–8 Hz, 8–10 Hz, 10–12 Hz, 12–30 Hz, 30–50 Hz.	Standard tests and imaging for diagnosis. Synchronization likelihood (coherence measure), comparing each channel with all other channels.	Negative correlation in the 10–12 Hz and 12–30 Hz bands between synchronization likelihood and age. Synchronization likelihood decreased in the upper alpha (10–12) and beta (12–30) bands in AD compared to SC. In the remaining condition, the synchronization likelihood was significantly higher in AD than in MCI in the 0.5–4 Hz frequency band. During the working memory task, the synchronization likelihood was significantly higher in MCI compared to the SC in the lower alpha band (8–10 Hz).
Prichep et al., 2006 [83]		Theta (3.5–7.5 Hz).	Multiple logistic regression	Multiple logistic regression of theta power (3.5–7.5 Hz), mean frequency, and interhemispheric coherence predicted the decline from MCI to AD at long term with an overall predictive accuracy of about 90%.
van der Hiele et al., 2007 [69]	22 HC, 18 MCI and 16 probable AD	Theta (4– 8 Hz) and alpha (8–13 Hz).	Standardized test. Spectral analysis. Theta relative power (% theta in the 4–13 Hz band), alpha reactivity and alpha spectral coherence during eyes closed and memory activation. EEG power measures averaged over all electrode positions.	Theta relative power (% of the 4-13 Hz) in AD > MCI > HC and related to decreased performance in all cognitive domains. Theta absolute power AD> HC. Alpha reactivity HC > AD and related to decreased performance on tests of global cognition, memory, and executive functioning.
Schreiter Gasser et al., 2008 [55]	54 AD, 24 mixed dementia (vascular Alzheimer) and 66 HC	Delta (1.5–3.5 Hz), theta (3.5–7.5 Hz), alpha1 (7.5–9.5 Hz), alpha2 (9.5–12.5 Hz), beta1 (12.5–18.5 Hz), beta2 (18.5–25.0 Hz).	Standard clinical and neuropsychological tests, neuroradiology (CT) and qEEG. Spectral power.	Neuroimaging and qEEG made a greater differential diagnostic contribution than clinical symptoms and neuropsychology. Delta power: Mixed > AD > HC. High frequency power decreased in AD. Topography of slow band changed for fast bands: both patient groups showed a flattening in the anterior–posterior distribution in alpha2, beta1, and beta2.
Babiloni et al., 2009a [85]	64 HC, 69 amnesic MCI, and 73 mild AD	Delta (2–4 Hz), theta (4–8 Hz), alpha 1 (8–10 Hz), alpha 2 (10–12 Hz), beta 1 (13–20 Hz), beta 2 (20–30 Hz), and gamma (30–40 Hz).	Standard clinical and neuropsychological tests. Neuroimaging (CT, MRI) and laboratory analyses. Direction of information flux within EEG functional coupling by directed transfer function (DTF) with Mvar model. EEG power density spectrum and relative power. Directionality between F3–P3, Fz–Pz, F4–P4. Interhemispheric directionality between F3–F4, C3–C4, P3–P4.	Parietal to frontal direction of the information flux within EEG functional coupling of theta: HC > MCI/AD; Alpha1: HC > MCI/AD; Alpha2: HC > MCI/AD; beta1: HC > MCI > AD; beta2: HC > AD. No differences in the directional flow within inter-hemispheric EEG functional coupling.
Babiloni et al., 2009b [118]	60 HC, 88 MCI, and 35 AD	Delta (2–4Hz), theta (4–8 Hz), alpha 1 (8–10.5 Hz), alpha 2 (10.5–13 Hz), beta 1 (13–20 Hz), and beta 2 (20–30 Hz).	Standard tests. Spectral analysis: power spectrum, alpha peak frequency. EEG sources by low resolution electromagnetic source tomography (LORETA).	Significant linear correlation of hippocampal volume with the magnitude of alpha1 sources in the parietal, occipital and temporal areas. Progressive atrophy of hippocampus correlates with decreased cortical alpha power in a continuum MCI < AD.
Gallego-Jutglà et al., 2012 [119]	24 HC (10 male, 14 female), 17 mild AD (9 male, 8 female)	Narrow frequency bands of different sizes, with the aim of optimizing the band selection.	Standard tests. Synchrony analysis by cross-correlation, phase synchrony and Granger causality.	The frequency range 5–6 Hz yields the best accuracy for diagnosing AD (within the classical theta band) for directed transfer function (DTF) Granger causality.
Babiloni et al., 2013 [62]	88 mild AD (19 male, 69 female), 35 HC (6 male, 29 female)	Delta (2–4 Hz), theta (4–8 Hz), alpha 1 (8–10.5 Hz), alpha 2 (10.5–13 Hz), beta 1 (13–20 Hz), beta 2 (20–30 Hz), and gamma (30–40 Hz).	Standard tests. Spectral analysis: power spectrum. EEG sources by low resolution electromagnetic source tomography (LORETA).	Mild AD had a power increase in widespread delta sources and by a power decrease in posterior alpha sources. In mild AD, the follow-up EEG recordings showed increased power of widespread delta sources as well as decreased power of widespread alpha and posterior beta 1 sources.
Poil et al., 2013 [138]	86 MCI (25 MCI converters to AD and 61 others)	Broad band signal. Delta (1–3 Hz),theta (4–7 Hz), alpha (8–13 Hz), beta (13–30 Hz), and gamma (30–45 Hz), alpha divided into three narrower bands.	Logistic regression. Large-scale data mining (177 biomarkers). Neurophysiological Biomarker Toolbox (http://www.nbtwiki.net/).	Multiple EEG biomarkers mainly related to activity in the beta frequency range (13–30 Hz) can predict conversion from MCI to AD in 2 years. By integrating six EEG biomarkers into a diagnostic index using log regression, the prediction improved, with a sensitivity of 88% and specificity of 82%, compared with a sensitivity of 64% and specificity of 62% of the best individual biomarker in this index (peak width of the dominant beta peak).
Lizio et al., 2016 [70]	127 AD and 121 HC	Delta (2–4 Hz), theta (4–8 Hz), alpha 1 (8–10.5 Hz), alpha 2 (10.5–13 Hz), beta 1 (13–20 Hz), beta 2 (20–30 Hz), and gamma (30–40 Hz).	LORETA. Ratio between parieto-occipital cortical sources of delta and low-frequency alpha rhythms.	The ratio offered 77.2% of sensitivity in the recognition of the AD individuals; 65% of specificity in the recognition of the Nold individuals; and 0.75 of area under the receiver-operating characteristic curve.
Babiloni et al., 2016 [76]	19 AD patients with dementia and 40 healthy elderly subjects.	Delta (2–4 Hz) and low-frequency alpha (8–10.5 Hz)	LORETA. Fluorodeoxyglucose positron emission tomography (PET) images.	AD group pointed to lower activity of low-frequency alpha sources and higher activity of delta sources which correlates positively with glucose hypometabolism in the cortical region of interest.
Hata et al., 2017 [78]	14 probable Alzheimer’s disease patients	Delta (2–4 Hz), theta (4–8 Hz), alpha1 (8–10 Hz), alpha2 (10–13 Hz), beta1 (13–20 Hz), and beta2 (20–30 Hz).	eLORETA: current source density (CSD) and lagged phase synchronization (LPS). Brain MRI, cerebrospinal fluid measurements, and neuropsychological assessments.	Patients with AD showed significant negative correlation between CSF Aβ42 concentration and the logarithms of CSD over the right temporal area in the theta band. Total tau concentration was negatively correlated with the LPS between the left frontal eye field and the right auditory area in the alpha-2 band in patients with AD.
Houmani et al., 2018 [97]	169 patients:SCI (n = 22), MCI (n = 58), AD (n = 49), Other pathologies (n = 40)	0.1–4 Hz (delta), 4–8 Hz (theta), 8–12 Hz (alpha), 12–30 Hz (beta), 30–100 Hz (gamma)	Epoch-based entropy (signal complexity) and bump models (EEG local synchrony)	Automatic discrimination of possible AD patients from SCI patients and from MCI or other pathologies. Accuracy 91.6% (specificity = 100%, sensitivity = 87.8%) Discriminating SCI patients from possible AD patients’ accuracy 81.8% to 88.8%.
Handayani et al., 2018 [112]	22 elderly subjects consisted of 10 MCI subjects and 12 healthy subjects	Delta (1–4 Hz), theta (4–7 Hz), alpha (7–13 Hz), and beta (13–30 Hz).	Coherence between each electrode pair measured for all frequency bands.Magnitude of phase synchrony expressed in the phase locking value (PLV).	Decrease in intrahemispheric and interhemispheric coherence especially in the beta band.Decrease in signal synchronization in some electrode pairs for the alpha band and on all electrode pairs for beta band.
Smailovic et al., 2018 [80]	Subjective cognitive decline (SCD; n=210), mild cognitive impairment (MCI; n=230) and AD (n=197)	Delta (1–3.5 Hz), theta (4–7.5 Hz), alpha (8–11.5 Hz) and beta (12–19.5 Hz)	qEEG, global field power (GFP) and global field synchronization (GFS), and CSF biomarkers	Decreased CSF Aβ42 correlated with increased theta and delta GFP. Increased p- and t-tau with decreased alpha and beta GFP. Decreased CSF Aβ42 and increased p- and t-tau associated with decreased GFS alpha and beta.
Koelewijn et al., 2019 [113]	Healthy young humans (N = 183) genotyped for APOE-e4. AD patients (N = 14) and age-matched controls (N = 11)	Delta: 1–4 Hz, Theta: 3–8 Hz, Alpha: 8–13 Hz, Beta-13–30 Hz, LowGamma: 40–60 Hz, and High-Gamma: 60–140 Hz.	Amplitude–amplitude connectivity of beamformer-derived oscillatory source signals, across six frequency bands and 90 AAL atlas brain areas.	Connectivity across alpha/beta increased in APOE-e4 in right-hemisphere, lateral parietal and precuneus of the default mode network. Hyperactivity in gamma. Hypoconnectivity in bilateral network in AD.

AD: Alzheimer’s disease patients; CT: computerized tomography; HC: healthy controls; MCI: mild cognitive impairment patients; EEG electroencephalogram; qEEG: quantitative EEG.

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
