# Peer review of "Electroencephalography as a Non-Invasive Biomarker of Alzheimer’s Disease: A Forgotten Candidate to Substitute CSF Molecules?"

_ijms, 2021, doi:10.3390/ijms221910889_

Round 1
Reviewer 1 Report
- 1 : change to 1. There is no need to have .1. This section is summarized the past/current biomarker usages for AD.
- Page 2, line 88-90. Authors mentioned as “some of them (PET or MRI) are expensive to be used in routine clinical care and their availability could be limited in some hospitals.” Compere to that, can authors provide how much EEG test cost and whether it can be routine tests and how easy to access this test?
- Page 3, lines 103-104: “set of proteins that correlates with amyloid burden”. Authors need to provide more specific information. Provide some examples of “set of proteins” form reference (Monllor et al.). Do not expect a reader will check the reference for this.
- Page 3, line 111; ultrasensitive methods? Can authors describe more specific about this method to help a reader to understand what it is (without checking a reference)?
- Page 10, lines 375-376: Is EEG reliable method for predicting MCI to AD progression? describe what the references (108, 109) found.
- Page 11, line 403: delete “.” (before Conclusions)
- There are some mistakes in references. Delete ().
- From beginning, it clearly mentions the purpose of this review in abstract, as “This review aims to focus on the usefulness and limitations of electro-encephalography (EEG) in the search for Alzheimer’s disease (AD) biomarkers. “, however, I don’t clearly see what authors suggested. Especially, authors need to more clearly describe limitation of using EEG as biomarker for AD progression. And in conclusions (lines 412-414, pages 11), it would be better if authors provide how can overcome these differences in methodological approaches.
- Page 4, line 190; is “and” right in that sentence?
Author Response
Dear Editor of IJMS,
Find enclosed the revised version of our manuscript “Electroencephalography as a non-invasive biomarker of Alzheimer’s disease: a forgotten candidate to substitute CSF molecules?”. We thank the reviewers for their feedback, their comments have helped us substantially to improve our review.
For a detailed description please see below the changes that have been carried out to address each point the reviewers had raised.
We hope that you find now that this paper is of sufficient interest to warrant publication in International Journal of Molecular Science and look forward to receiving your news.
Yours sincerely
Ana Lloret
REVIEWER 1
- 1 : change to 1. There is no need to have .1. This section is summarized the past/current biomarker usages for AD.
We thank the reviewer for noticing the mistake. We have corrected it.
- Page 2, line 88-90. Authors mentioned as “some of them (PET or MRI) are expensive to be used in routine clinical care and their availability could be limited in some hospitals.” Compere to that, can authors provide how much EEG test cost and whether it can be routine tests and how easy to access this test?
We thank the reviewer for proposing this topic. We have found a comparative study perform by Dellabadia et al., and they found that MRI was three and PET was six times the cost of a EEG [143]. Moreover, the price of the MRI/PET equipment is, at least, seven times higher than a portable EEG. Finally, the necessary infrastructure is not comparable. The rooms where the resonance equipment is installed must have a special set-up at a high cost. Among other things, they require radio-frequency shielding. On the other hand, PET requires all the design and precautions of a radioactive facility. Floors, ceilings, walls, ventilation and climatization must be special. In addition, the radioactive material requires a special preparation process in cyclotrons and its disposal and handling also requires special knowledge and treatment. However, EEG is safe, clean, harmless, and can be performed in any environment. It does not need to be performed in hospitals, it can be performed in primary care centers or even in outpatient settings [144].
We have added a new paragraph explaining this topic (Pge 14, Lines 437-441) .
- Page 3, lines 103-104: “set of proteins that correlates with amyloid burden”. Authors need to provide more specific information. Provide some examples of “set of proteins” form reference (Monllor et al.). Do not expect a reader will check the reference for this.
We agree with the reviewer that reading is facilitated by providing these data. We have added the following sentence (page 3, lines 104-106):
“The levels of serum clusterin, PKR, and RAGE were statistically different in the AD group compared to controls and correlated with those of CSF Aβ42.”
- Page 3, line 111; ultrasensitive methods? Can authors describe more specific about this method to help a reader to understand what it is (without checking a reference)?
We thank the reviewer for the comment which will clarify information. We have changed “ultrasensitive” by “highly sensitive and specific immuno- and mass spectrometry-based assays” (Page 3, Lines 112-113).
- Page 10, lines 375-376: Is EEG reliable method for predicting MCI to AD progression? describe what the references (108, 109) found.
We have added the next paragraph (Page 13, Lines 402-408):
“Vecchio et al. [134] showed that EEG connectivity analysis, combined with a neuropsychological MCI pattern and ApoE genotyping reached high sensitivity/specificity and good classification accurate classification on an individual basis (>0.97 of AUC), helping to determine the risk of the progression to AD in MCI patients. Jelic et al. [135] found that the most important predictors were alpha and theta relative power and, as well as mean frequency from left temporo-occipital derivation (T5-O1), which classified 85% of MCI subjects correctly”.
- Page 11, line 403: delete “.” (before Conclusions)
We want to thank for noticing the mistake. We have now corrected it.
- There are some mistakes in references. Delete ().
We apologize for the mistakes which has been solved.
- From beginning, it clearly mentions the purpose of this review in abstract, as “This review aims to focus on the usefulness and limitations of electro-encephalography (EEG) in the search for Alzheimer’s disease (AD) biomarkers. “, however, I don’t clearly see what authors suggested. Especially, authors need to more clearly describe limitation of using EEG as biomarker for AD progression. And in conclusions (lines 412-414, pages 11), it would be better if authors provide how can overcome these differences in methodological approaches.
We thank the reviewer for this clarification. We have added the following paragraph expanding the information on limitations (Page 14, Lines 449-456):
“Currently, the International Federation of Clinical Neurophysiology is working on establishing unanimous recommendations for the topographic and frequency analysis of resting state EEGs. These recommendations included: how to record EEGs (environment, montage, settings); digital storage of EEG and control data; extraction of synchronization, connectivity and topographic features; and statistical analysis and neurophysiological interpretation of those EEG features [145]. Setting up an international working group specialized in dementias could be an interesting and useful idea in order to standardize the use of EEG in the diagnosis of EA.”
- Page 4, line 190; is “and” right in that sentence?
In order to clarify the sentence, we have changed it by other and we apologize for the possible confusion (Page 4, Lines 195-196):
“Additionally, it is important to consider not only the rhythms themselves, but also the brain area where these oscillations are recorded”.

Reviewer 2 Report
The manuscript “Electroencephalography as a non-invasive biomarker of Alzheimer’s disease: a forgotten candidate to substitute CSF molecules?”, by Monllor P, Cervera-Ferri A, Lloret M-A, Esteve D, Lopez B, Leon J-L and Lloret A, is a very interesting review about a highly relevant topic in the field of Alzheimer’s disease (AD) and neurophysiology. Authors have done a very well work, although there are some questions that must be clarified before a final publication.
1.- There are several inaccurate sentences and affirmations.
1.1.- Line 127-128. It is stated that interactions between neuronal populations generate electrical fluxes, but it is just the opposite way, in fact are the electric currents, which modify neuronal activity, the responsible for the interactions. Besides, electric flux is the measure of the electric field through a given surface, but this magnitude cannot be measured in neurophysiology. Modify this sentence in a more rigorous sense.
1.2.- Line 146. “Brain oscillations”. Brain does not oscillate, except for heartbeat. See also line 267. Please, be more rigorous.
1.3.- Line 151. “Oscillatory frequencies”. What are oscillatory frequencies? Non- stationary frequencies changing periodically over time?. Please, clarify.
1.4.- Line 154. The affirmation that “fast oscillations reflect(ing) local spikes” is misleading and inaccurate. Please, modify this error.
1.5.- Line 266. “Spectral analysis does not reflect the time evolution of signals”. Obviously, this sentence is not rigth, because you can obtain a dynamical picture of the system with sequential spectral analysis. Moreover, the same can be said from network analysis or any other method to study synchronizity or complexity, taking into account that all of them use windowing.
1.6.- Lines 269-270. “Therefore, the EEG exhibits complex behaviors which cannot be linearly analyzed”. This sentence is also misleading because not all the interactions in the brain are non-linear and, in fact, a lot of brain signals are linear (e.g, see Ortega et al, Epilepsia, 2008 49(2):269–280; Ortega et al, Clin. Neurophysiol. 122 (2011) 1106–1116).
1.7.- Lines 304-305. Definition of EEG synchrony. Although this definition of synchrony is correct for a visual description of EEG, obviously is useless for a numerical analysis. Please, see Pikovsky, A., Rosenblum, M. & Kurths, J. (2001). Synchronization: A Universal Concept in
Nonlinear Sciences. Cambridge Nonlinear Science Series.
2.- Considering that all the reviewed papers use qEEG, they should include a small section on numerical analysis, at least as Appendix. A possible reference could be
Necessity of quantitative EEG in daily clinical practice. Pastor et al. In: Electroencephalography. Ed. Hideki Nakano, InTech 2021.
3.- LORETA is a method to approximate a solution to the inverse problem (what is impossible theoretically!) but not for study the frequency pattern of EEG. Clarify this aspect.
4.- The table at pages 5-8 is not properly referenced at text and there is neither title nor introductory heading.
5.- The paragraph entitled The EEG as a biomarker of AD should be changed as Discussion, because it sounds repetitive.
Author Response
Dear Editor of IJMS,
Find enclosed the revised version of our manuscript “Electroencephalography as a non-invasive biomarker of Alzheimer’s disease: a forgotten candidate to substitute CSF molecules?”. We thank the reviewers for their feedback, their comments have helped us substantially to improve our review.
For a detailed description please see below the changes that have been carried out to address each point the reviewers had raised.
We hope that you find now that this paper is of sufficient interest to warrant publication in International Journal of Molecular Science and look forward to receiving your news.
Yours sincerely
Ana Lloret
REVIEWER 2
The manuscript “Electroencephalography as a non-invasive biomarker of Alzheimer’s disease: a forgotten candidate to substitute CSF molecules?”, by Monllor P, Cervera-Ferri A, Lloret M-A, Esteve D, Lopez B, Leon J-L and Lloret A, is a very interesting review about a highly relevant topic in the field of Alzheimer’s disease (AD) and neurophysiology. Authors have done a very well work, although there are some questions that must be clarified before a final publication.
1.- There are several inaccurate sentences and affirmations
We appreciate the reviewer’s positive comment and his/her feedback about the manuscript. Please find below our point-by-point answer to his/her remarks.
1.1.- Line 127-128. It is stated that interactions between neuronal populations generate electrical fluxes, but it is just the opposite way, in fact are the electric currents, which modify neuronal activity, the responsible for the interactions. Besides, electric flux is the measure of the electric field through a given surface, but this magnitude cannot be measured in neurophysiology. Modify this sentence in a more rigorous sense.
We appreciate the reviewer’s consideration. We have rewritten the text as (Page 3, Lines 130-133):
“Such interactions involve electrical fluxes, which are reflected in voltage fluctuations that can be measured invasively at the cellular level (intra- or extracellular neuronal spike recordings), at a local network level (local field potentials) or even at longer distances, at cortical levels (electrocorticograms)”.
1.2.- Line 146. “Brain oscillations”. Brain does not oscillate, except for heartbeat. See also line 267. Please, be more rigorous.
We have changed the expression “brain oscillations” it to “neural oscillations” or “brain rhythms” in the manuscript.
1.3.- Line 151. “Oscillatory frequencies”. What are oscillatory frequencies? Non- stationary frequencies changing periodically over time?. Please, clarify.
To clarify, “Oscillatory frequencies” have been replaced by “The frequency of the oscillations”.
1.4.- Line 154. The affirmation that “fast oscillations reflect(ing) local spikes” is misleading and inaccurate. Please, modify this error.
We have modified the text, as follows (Page 4, Lines 154-164):
“The frequency of the oscillations relates to the extent of the network involved and is impacted by the axonal conduction velocity [37, 38]: while neighboring neurons can communicate very fast, through the use of low-amplitude fast oscillations, distant regions need slower oscillations, which recruit bigger neuronal populations and, therefore, exhibit higher amplitude. At cortical levels, interneurons usually resonate at gamma frequencies [39-43] and therefore their interactions with pyramidal cells can induce local cycles of excitation and inhibition at these frequencies [44]. Therefore, fast oscillations emerge from these local interactions. In contrast, low-frequency oscillations can be recorded at longer distances. These slow waves are travelling waves [45-47], and thus, can be recorded far from their oscillatory generators and reflect wider network interactions at an interregional level [48].”
1.5.- Line 266. “Spectral analysis does not reflect the time evolution of signals”. Obviously, this sentence is not rigth, because you can obtain a dynamical picture of the system with sequential spectral analysis. Moreover, the same can be said from network analysis or any other method to study synchronizity or complexity, taking into account that all of them use windowing.
The text has been deleted. We understand that sequential spectral analysis can provide useful information about the temporal evolution of the EEG. Also, spectral analyses are less resource-consuming for computation than time-frequency analyses and, thus, can be used successfully in the clinical practice. However, given the signals are non-stationary because they change with time, there has been much research for improving time-frequency analysis. Conceptually, spectral analysis only decomposes the signals into its frequency components, and so the time component is not considered per se in the decomposition. Thus, we can miss relevant changes involving the temporal evolution of the signals when only spectral analyses are used if these changes occur very fast for the chosen windows.
1.6.- Lines 269-270. “Therefore, the EEG exhibits complex behaviors which cannot be linearly analyzed”. This sentence is also misleading because not all the interactions in the brain are non-linear and, in fact, a lot of brain signals are linear (e.g, see Ortega et al, Epilepsia, 2008 49(2):269–280; Ortega et al, Clin. Neurophysiol. 122 (2011) 1106–1116).
The text has been changed (Page 11, Lines 289-294):
“EEG signals are non-stationary, complex and non-linear signals [87], caused by the interaction of different sources (oscillators) [49]. Therefore, the EEG exhibits complex behaviors which cannot be linearly analyzed, and consequently, non-linear analyses have been introduced to the study of EEG signals to quantify this complexity [88]. In some diseases, the combination of linear and non-linear analyses can improve the accuracy EEG-derived biomarkers [87].”
1.7.- Lines 304-305. Definition of EEG synchrony. Although this definition of synchrony is correct for a visual description of EEG, obviously is useless for a numerical analysis. Please, see Pikovsky, A., Rosenblum, M. & Kurths, J. (2001). Synchronization: A Universal Concept in Nonlinear Sciences. Cambridge Nonlinear Science Series.
We thank the reviewer for the bibliography recommendation, it is really interesting for us. The text has been replaced by (Page 11, Lines 327-331):
“Synchronization between neuronal populations is relevant for to the interaction among neural networks. EEG synchrony refers to the adjustment of different neural oscillations. Two different signals become coupled when both of them begin oscillating at the same frequency, become phase-locked, experience phase-amplitude coupling or modulate their amplitudes together (amplitude-amplitude coupling) [104].”
2.- Considering that all the reviewed papers use qEEG, they should include a small section on numerical analysis, at least as Appendix. A possible reference could be Necessity of quantitative EEG in daily clinical practice. Pastor et al. In: Electroencephalography. Ed. Hideki Nakano, InTech 2021.
Again, we thank the reviewer for his/her contribution with this recommendation. We agree that qEEG is needed in clinical practice in order to avoid the subjectivity in the interpretation. Since the specific detail on the different measures perhaps exceeds the aims of the review, we have included a new paragraph in section 3 (The EEG as a result of brain activity) as a remark of the needing of qEEG analysis (Pages 4-5, Lines 199-207).
“Finally, although the EEG provides immediate reports of the brain’s electrical activity, this technique is usually restricted in clinical practice to study epilepsy and sleep disorders, and is sometimes considered as a subjective technique [57]. This last consideration is due to excessive de visu interpretation of data, which could lead to a high variability in results. In order to make an objective analysis, the use of numerical analysis of the signals (quantitative EEG or qEEG) is strongly encouraged [58]. The focus of this review lies on the results of qEEG analysis conducted on AD patients. These analyses mainly involve spectral decomposition of the signals into their frequency components and the study of their synchronization and complexity (see Table 1).”
3.- LORETA is a method to approximate a solution to the inverse problem (what is impossible theoretically!) but not for study the frequency pattern of EEG. Clarify this aspect.
We thank the reviewer for his/her observation. We have included a brief explanation about LORETA to make it clearer (Page 5, Lines 220-223):
“Babiloni et al. used low-resolution brain electromagnetic tomography (LORETA) analysis in a large sample of healthy elderly and young individuals. By combining imaging and qEEG analysis, this technique estimates the location of electrical activity generators in the brain, which is also named as the inverse problem [59].”
4.- The table at pages 5-8 is not properly referenced at text and there is neither title nor introductory heading.
We thank the reviewer for noticing these mistakes. Table 1 has been referenced and a heading has been added.
5.- The paragraph entitled The EEG as a biomarker of AD should be changed as Discussion, because it sounds repetitive.
The title has been changed.

Round 2
Reviewer 1 Report
Summary:
The authors review key features of EEG and changes in AD, namely slowing of EEG (reduction of fast and increase of slow frequential components), perturbation in EEG synchrony and reduction in complexity measures. This review is an important contribution to the continuous development efforts of non-invasive and cost-effective biomarkers for diagnosis and disease progression monitoring of AD.
Broad comments:
As the authors state, reviewing of the EEG literature should focus on actual frequencies reported and not just classification of frequencies, as there is a lack of standardized classifications. Similarly, studies that have taken an integrative approach (Poil et al.), extracting multiple EEG biomarkers, are particularly useful for the development of EEG as an AD biomarker.
Furthermore, a key point that could be expanded upon in the text and again be reemphasized in the Conclusion section, is that any other biomarker measurement available besides for cognitive status in the literature reviewed should be mentioned. A few studies are brought up that explores the relationship to CSF and brain volume (Section 4.1). However, if any of the studies mentioned includes any other biomarker change besides cognitive status, or otherwise at least the cognitive tests used for diagnosis, that would have been helpful to include in the table, given the challenges of diagnosing and characterizing AD, and developing biomarkers.
Section 4.1 Change of frequency pattern on EEG second paragraph: Changes in EEG with normal aging could be expanded upon further, as normal aging leads to changes itself and aging age-matched controls are the control groups used for AD. For example, the comparison of EEG changes to other dementias is well-described.
Specific comments:
Row 54-55: Tangles consists of tau protein and does not include the dying cells to my understanding.
Row 61: Perhaps use completely established rather than totally.
Author Response
Summary:
The authors review key features of EEG and changes in AD, namely slowing of EEG (reduction of fast and increase of slow frequential components), perturbation in EEG synchrony and reduction in complexity measures. This review is an important contribution to the continuous development efforts of non-invasive and cost-effective biomarkers for diagnosis and disease progression monitoring of AD.
We want to thank the reviewer for his/her comments
Broad comments:
As the authors state, reviewing of the EEG literature should focus on actual frequencies reported and not just classification of frequencies, as there is a lack of standardized classifications. Similarly, studies that have taken an integrative approach (Poil et al.), extracting multiple EEG biomarkers, are particularly useful for the development of EEG as an AD biomarker.
Yes, we agree with the reviewer on the importance of standardized classifications and searching for multiple EEG biomarkers
Furthermore, a key point that could be expanded upon in the text and again be reemphasized in the Conclusion section, is that any other biomarker measurement available besides for cognitive status in the literature reviewed should be mentioned. A few studies are brought up that explores the relationship to CSF and brain volume (Section 4.1). However, if any of the studies mentioned includes any other biomarker change besides cognitive status, or otherwise at least the cognitive tests used for diagnosis, that would have been helpful to include in the table, given the challenges of diagnosing and characterizing AD, and developing biomarkers.
We thank the reviewer for this pertinent comment. We have added some new references in 4.1 section and included them in the table.
“Importantly, this slowing of EEG correlates with gold standard biomarkers of AD, such as Fluorodeoxyglucose-PET images [79] and phosphorylated tau and/or Aβ42 measured in CSF [80-83].”
Section 4.1 Change of frequency pattern on EEG second paragraph: Changes in EEG with normal aging could be expanded upon further, as normal aging leads to changes itself and aging age-matched controls are the control groups used for AD. For example, the comparison of EEG changes to other dementias is well-described.
We have added the next paragraph in Section 4:
“Healthy aging is the cause of changes in the activity of the brain that, accordingly, are reflected in EEG recordings. In summary, they include a reduction in the amplitude of alpha activity (8–13 Hz), the slowing of the background activity, and an increase of delta (1–4 Hz) and theta (4–8 Hz) power [59]. However, all these physiological EEG variations are pathologically exacerbated in AD patients. Next, we will analyze the main EEG anomalies found in Alzheimer's patients. They can be grouped into three categories: change of frequency pattern, reduction of the complexity of EEG signals and perturbation in EEG synchrony.”
And this paragraph in the conclusions section:
“Although EEG recordings in the elderly are different compared to those in younger people, the process of dementia leads to the appearance of pathological changes in the EEG that are clearly distinguishable from those of aged-matched controls.”
Specific comments:
Row 54-55: Tangles consists of tau protein and does not include the dying cells to my understanding. Row 61: Perhaps use completely established rather than totally.
We have changed these two sentences as has been recommended: “and the formation of neurofibrillary tangles composed mainly of abnormally phosphorylated tau (p-tau) protein”

Reviewer 2 Report
The manuscript has been substantially improved, but it remains some misleading expressions that must be corrected, because a precise terminology is extremely important in science.
1.- There are several inaccurate sentences and affirmations.
1.1.- Line 130-133.
A flux ([J]=mol/cm2 s) is defined as (see below), where is the surface across particles are moving through and n is the mass of a substance (mol). Obviously, if these are charged particles, the flux is an electric current, but not an electric flux, concept defined as the magnitude of electric field ( ) going through a surface (S), according to this expression (see Plonsey, Collin, Principles and Applications of Electromagnetic Fields, McGraw-Hill 1961 or Wagness, Campos Electromagnéticos, Limusa, 2001)
See equation at file
Where is the vector surface element.
What the authors mean by electrical fluxes are transmembrane currents, please use this more appropriate and precise concept.
Please, see Word doc attached

Author Response
The manuscript has been substantially improved, but it remains some misleading expressions that must be corrected, because a precise terminology is extremely important in science.
1.- There are several inaccurate sentences and affirmations.
1.1.- Line 130-133.
A flux ([J]=mol/cm2 s) is defined as (see below), where is the surface across particles are moving through and n is the mass of a substance (mol). Obviously, if these are charged particles, the flux is an electric current, but not an electric flux, concept defined as the magnitude of electric field ( ) going through a surface (S), according to this expression (see Plonsey, Collin, Principles and Applications of Electromagnetic Fields, McGraw-Hill 1961 or Wagness, Campos Electromagnéticos, Limusa, 2001)
See equation at file. Where is the vector surface element. What the authors mean by electrical fluxes are transmembrane currents, please use this more appropriate and precise concept.
We sincerely thank the reviewer by his/her contribution and clarification, and apologize for the mistake. The text has been changed according to the suggestion. The expression “electrical fluxes” has been substituted by “transmembrane currents” in Section 3.
